# Uncertainty in ensembles of global biodiversity scenarios

Wilfried Thuiller[1], Maya Guéguen[1], Julien Renaud[1], Dirk N. Karger[2] & Niklaus E. Zimmermann [2]

While there is a clear demand for scenarios that provide alternative states in biodiversity with respect to future emissions, a thorough analysis and communication of the associated uncertainties is still missing. Here, we modelled the global distribution of ~11,500 amphibian, bird and mammal species and project their climatic suitability into the time horizon 2050 and 2070, while varying the input data used. By this, we explore the uncertainties originating from selecting species distribution models (SDMs), dispersal strategies, global circulation models (GCMs), and representative concentration pathways (RCPs). We demonstrate the overwhelming influence of SDMs and RCPs on future biodiversity projections, followed by dispersal strategies and GCMs. The relative importance of each component varies in space but also with the selected sensitivity metrics and with species' range size. Overall, this means using multiple SDMs, RCPs, dispersal assumptions and GCMs is a necessity in any biodiversity scenario assessment, to explicitly report associated uncertainties.

[1] Univ. Grenoble Alpes, Univ. Savoie Mont Blanc, CNRS, LECA, Laboratoire d'Écologie Alpine, F- 38000 Grenoble, France. [2] Swiss Federal Research Institute WSL, 8903 Birmensdorf, Switzerland. Correspondence and requests for materials should be addressed to W.T. (email: wilfried.thuiller@univ-grenoble-alpes.fr)

The launch of the Intergovernmental Science-Policy Platform on Biodiversity and Ecosystem Services (IPBES) in 2012 and the ongoing activities of the Intergovernmental Panel for Climate Change (IPCC) have led the ecological communities to conceive and produce biodiversity models and scenarios[1,2]. We define here biodiversity models as any type of quantitative model that aims at predicting the spatial or/and temporal distribution of a population, species or groups of species. Biodiversity scenarios, the application of these models to plausible trajectories of different aspects of the future (e.g. future climate), rely on several complex components that depend on each other (Fig. 1 in ref. [2]). When focusing on biodiversity trends under climate change, biodiversity models are first trained or fitted under current climatic conditions.

**Fig. 1** Relative influence of the different sources of variation on species' level sensitivity metrics. The plot represents the relative influence of the choice of species distribution models (SDMs), global circulation models (GCMs) and representative concentration pathways (RCPs) on the variance of change in climatic suitability (left) and loss of suitable climate (right) for all modelled species. The deviance was calculated across all species by means of a nested ANOVA and the partitioning is represented by the percentage of explained deviance (note that the unexplained deviance is not represented here). ANOVAs were run for three different TSS thresholds (0.4, 0.6, and 0.7) above which individual SDMs were retained for assessing the model results

Next, these models project future biodiversity patterns based on climate projections from global circulation models (GCMs) that in turn rely on socio-economic scenarios, expressed as representative concentration pathways (RCPs[3]). This chain of dependencies implies that the coupled choice of a GCM and a RCP is not straightforward[4]. Analyzing the biodiversity response with respect to RCPs is crucial to test and report how climate change and its mitigation could ultimately influence biodiversity, with consequences for management and conservation. However, the variation among climate change projections from different GCMs can be so high that the interest of choosing one RCP over another is no longer seen a necessity[4]. With regards to biodiversity impact modelling, the situation is even more complex since biodiversity models themselves proved to generate considerable variability[5,6]. In other words, the choice of a biodiversity model, a GCM and a RCP matters and the uncertainties originating from these choices should be considered in decision-making or biodiversity conservation. In both the climate and ecological modelling communities, researchers are generating and increasingly using ensembles of projections from a range of GCMs, biodiversity models and RCPs to quantify the uncertainty arising from these apparently subjective choices[7,8]. However, while several studies have highlighted the potential consequences of ignoring this variability originating from different choices in biodiversity scenarios[9,10], most studies still produce single projections based on a single biodiversity model and a single GCM and only superficially discuss their results in the light of socio-economic pathways (see e.g.[11]).

Here, we demonstrate the importance of considering multiple approaches (e.g. biodiversity models, GCMs and RCPs) but also multiple ecological decisions (e.g. quality of the data, species dispersal limitation) on biodiversity scenarios. To obtain plausible and robust results useful for biodiversity assessments like IPBES[12], we run an extensive climate change impact assessment study by modelling the current and future climatic suitability of ~11,500 vertebrate species at the global scale (Supplementary Fig. 1). More specifically, we model the distribution of 1351 amphibian, 7248 bird, and 2896 mammal species as a function of current climate using four species distribution models (SDM) that are cross-validated four times and projected their future climatic suitability as a function of 14 combinations of GCMs and RCPs and for two dispersal strategies ('no dispersal' and 'limited dispersal'). This totaled to five million single projections from which we define two species-level sensitivity metrics (change in climatic suitability and loss in climatic suitability better) and pixel-level variation in community metrics (change in species richness, change in spatial turnover per region and temporal turnover per pixel). We then use nested ANOVAs to disentangle the relative importance of the choices of dispersal strategies, SDMs, GCMs and RCPs on the variability of the results.

## Results

**Species-level uncertainty analyses**. At the species-level, the influence of SDM, GCM and RCP differed with respect to the sensitivity metric and dispersal scenario used (Fig. 1). When assuming limited dispersal, the choice of SDMs causes a more than ten times higher deviance in the change of suitable habitats than the choice of the other two components. In other words, the subjective selection of a specific SDM has an overarching influence on the final result compared to the choices of GCMs and RCPs (Fig. 1). Interestingly, when considering only the loss in climatic suitability ('no dispersal' scenario), RCPs are the most important element causing variation, followed by SDMs and GCMs (Fig. 1). The observed difference in the explained

deviance between the two dispersal assumptions arises from the discrepancy in SDMs in projecting the future climatic suitability of species[13]. While SDMs generally agree well when predicting the current distribution of species (as is targeted by the no dispersal assumption, Supplementary Fig. 2), they are known to widely differ when projecting future distributions[14]. This effect is evident even when keeping only SDMs that reach very high predictive accuracies (TSS > 0.7, Fig. 1). In other words, it is not only the quality of the SDMs that explains a large proportion of variance in model output, but also their internal structure. Indeed, relatively complex models (i.e. random forest) use a variety of combinations of features that can lead to similar geographical predictions under current conditions, but that can vastly differ to other models when transferred to future conditions, especially under conditions that have been used for calibrating the modes. Some of the particular combinations of features could represent true complex interactions between species' occurrences and environmental conditions, but sometimes they could result of over-fitting and spurious combinations, that once projected in new conditions lead to misleading results. The ensemble projection of the bearded woodpecker (*Dendropicos namaquus*, Lichtenstein 1793) exemplifies this aspect (Supplementary Fig. 3). The results reveal marked differences for SDM in projected future climatic suitability which leads to difficulties when interpreting the results in the light of socio-economic pathways. On the other hand, the loss in (current) climatic suitability does not vary as much for SDM and rather show strong differences for RCP. Novel conditions are usually far away from the current range of species, which also explain that SDMs are a more important driver of over-all uncertainty when dispersal in accounted for, than when considering only current loss of suitable habitats. Interestingly, the same trend was also found to be a function of species' range size (Fig. 2). When considering dispersal, the effect of SDMs on total projection uncertainty was highest for rare species. However, when considering the loss of currently suitable climatic habitat (as is targeted by the no dispersal assumption), the influence of RCPs on projection uncertainty increases with species range sizes. For the rarest species, both SDMs and RCPs have the highest contributions to explain uncertainties, while for large-ranged species, the influence of RCPs on the total variance in future climatic suitability is largest. The same pattern emerges when considering the effects of selecting RCPs for a given GCM (Supplementary Fig. 4). Here, under limited dispersal, the deviance due to SDMs is such that there is no influence of RCPs on the final results. Only when considering the loss in climatic suitability (LCS), the effect of RCPs is discernible where the median LCS is around 65% under the RCP 8.5 and 30% under the RCP 2.6 (Supplementary Fig. 4). With regards to biodiversity management, our results indicate that only when considering the loss of currently suitable climate that one can assess the effects of climate change adaptation plans or of emission scenarios.

**Pixel-based uncertainty analyses**. The overall variation originating from the combination of SDMs, GCMs and RCPs markedly differed between pixel-based metrics and among species groups (Fig. 3). Under the 'no dispersal' assumption the amphibians showed lower variation than the other groups, while the variation in the other metrics was generally higher for amphibians. This is probably reflecting that amphibian species are slightly less easy to model (Supplementary Fig. 2) at that particular resolution (100 km) and scale (global) and generally occur in location where RCPs and GCMs also diverge most (Supplementary Figs. 5–8). Interestingly, the change in α-diversity was

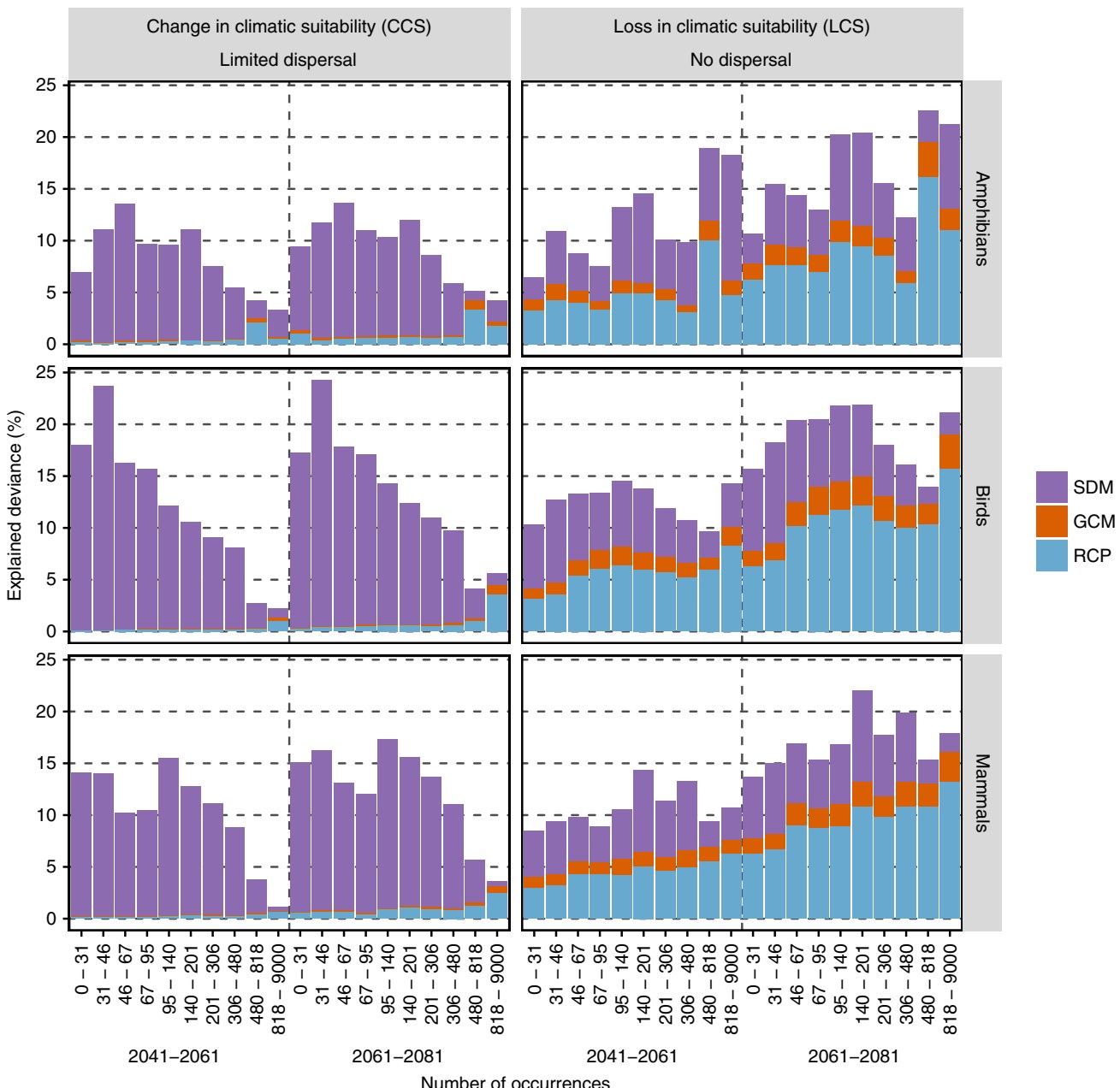

**Fig. 2** Relative influence of the different sources of variation on species' level sensitivity metrics in function of species range size. The plot represents the relative influence of SDMs, GCMs, and RCPs on overall species sensitivity to climate change in response to species' range size. A TSS threshold of 0.7 was used for all analyses. Species ranges were classified for equal size at the log scale

particularly sensitive to the choice of SDMs, which explained by far most of the deviance for that metric (Fig. 3). This was much less the case for the estimates of relative species turnover and loss per pixel were a larger proportion of the deviance was additionally explained by the RCPs, with the strongest effects for the time horizon 2070. While SDMs explained a constant portion of the deviance, the percentage of explained deviance by RCP scenarios increased considerably for 2070 (Fig. 3). This result confirms that when using sensitivity metrics that account for dispersal limitation (e.g. change in α-diversity), the uncertainty due to the choice of SDMs becomes particularly important and may hide variations due to RCPs.

Interestingly, the relative influence of SDMs, GCMs, and RCPs markedly varied in space at the scale of the IPBES sub-region (Fig. 4, Supplementary Fig. 9) but also at the pixel-scale

(Supplementary Figs. 10–15). This was particularly true for the effect of selecting GCMs, which on average have a low effect, but appear relatively strong in explaining the deviance in both %loss (percent of currently suitable area lost) and Δβs (percent change in total diversity per IPBES sub-region divided by the mean α-diversity per sub-region), under 'no dispersal' in all of the Europe and Central Asia region, and in the North Africa and the Australia sub-regions for birds, and to a lesser extent for mammals and amphibians (Fig. 4). Similarly, the deviance explained by RCPs for βt (temporal turnover) was much higher than the one explained by SDMs in Africa and South America, while this was opposite in Europe, Central Asia and North America (Fig. 4). These varying effects can be explained by the spatial structure of the predicted changes in the different sensitivity metrics (Supplementary Figs. 10–15). For instance,

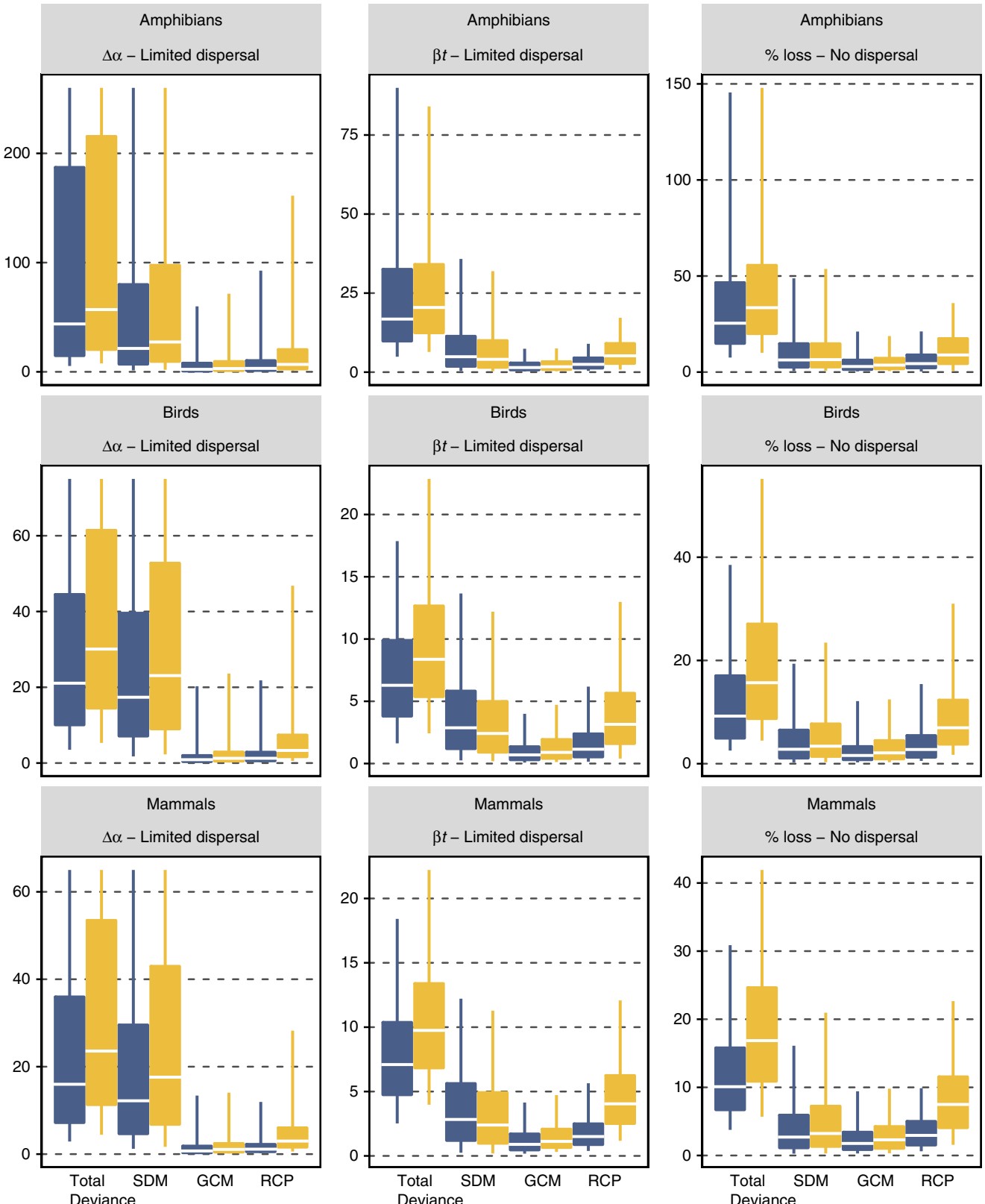

**Fig. 3** Absolute influence of the different sources of variation on pixel-based sensitivity metrics. The plot represents the absolute influence of the choice of species distribution models (SDMs), global circulation models (GCMs) and representative concentration pathways (RCPs) on the change in α-diversity per-pixel (Δα), temporal species turnover (βt) and percentage of species loss per pixel (% loss). The deviance was calculated across all pixels together by means of a nested ANOVA and the partitioning is represented by the absolute deviance to show the difference in deviance between metrics. Dark and light grey correspond to the horizon 2041–2060 and 2061–2080, respectively. Total deviance bar shows the total deviance that was explained by all components in the ANOVA. The central line of each box correspond to the median, the bounds of box represent the 25 and 75% quantiles, and the whiskers represent the quantiles 0.05 and 95%

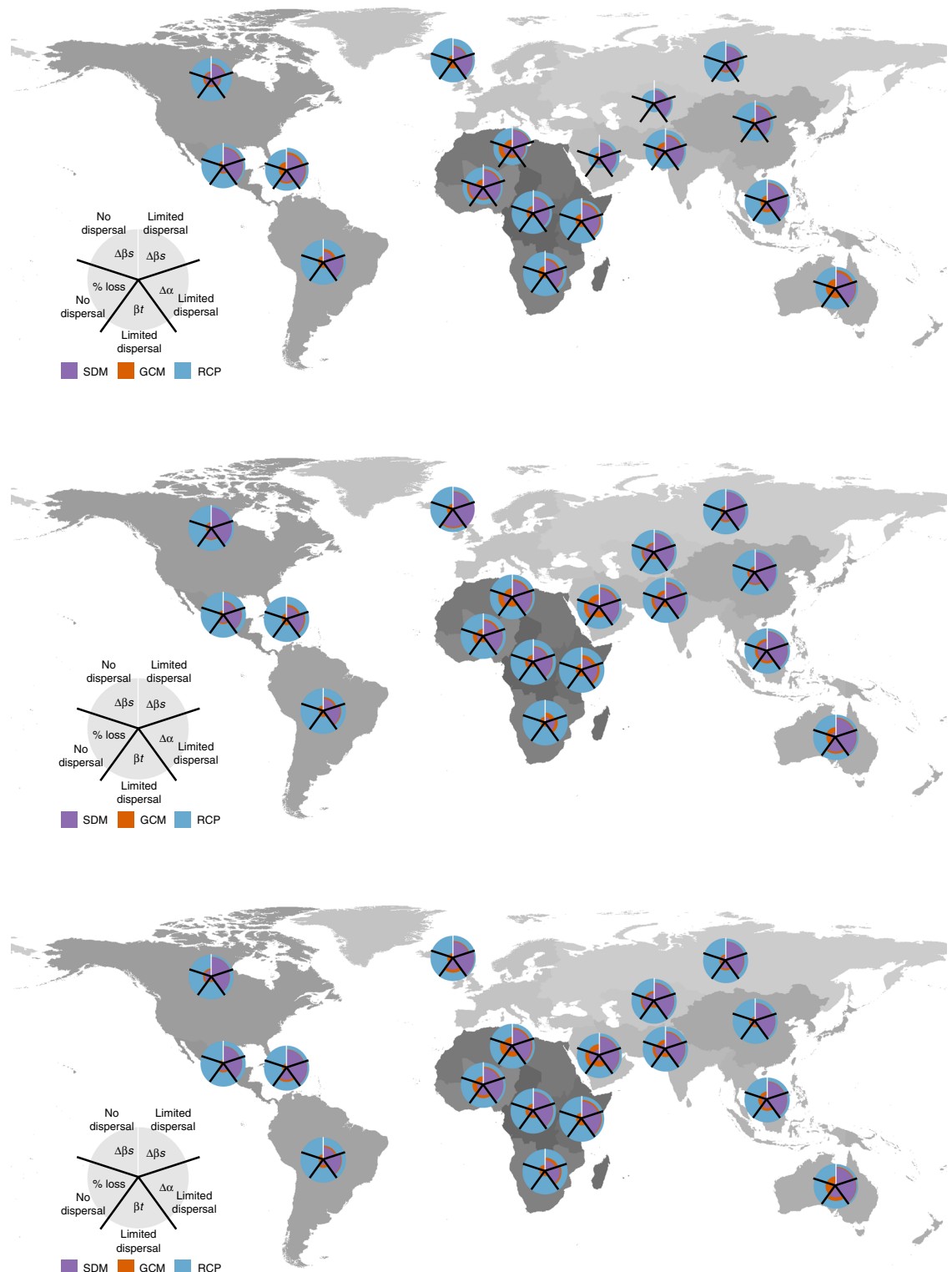

**Fig. 4** Spatial variation of the relative influence of the different sources of variation. The maps represent the relative influence (i.e. % of explained variance) of the choice of species distribution models (SDMs), global circulation models (GCMs) and representative concentration pathways (RCPs) on the change in α-diversity per-pixel (Δα), temporal species turnover (βt), percentage of species loss per pixel and change in β-diversity per IPBES sub-region (Δβs). Results are for the time period 2061–2080 (see Supplementary Fig. 9 for the period 2041–2060). Top row corresponds to amphibians, middle row to birds, and bottom-row to mammals

the areas with the highest projected changes in the percentage of species loss by 2070 were mostly located in Africa and South America, areas where most of the variation was explained by RCPs (Supplementary Fig. 12). A similar result was obtained for changes in α-diversity (Supplementary Fig. 11), where the areas with the highest expected changes (Northern Americas, Northern Africa, some areas in Central Asia) coincide with areas for which the largest deviance was explained by RCPs again.

## Discussion

In our comprehensive study, we have demonstrated the importance of considering multiple biodiversity models, multiple GCM, and, of course, multiple emission scenarios. Here, by covering the global scale and a larger range of organisms, we also demonstrate how the uncertainty in sensitivity metrics varies in space, in time, as a function of the organism, and ultimately, as a function of the sensitivity metric used. When considering sensitivity metrics that account for species dispersal capacity, the influence of the modelling algorithms becomes overwhelming. Statistical SDMs are known to markedly differ when projecting species distributions across space and time and this is why several packages include multiple algorithms to explore this variation (e.g.[15]). Importantly, even when selecting only the best performing models (TSS > 0.7), SDMs still caused highest contributions to uncertainty, followed by GCMs, and then RCPs.

For biodiversity management, this could have important implications. There is indeed growing interest in climate change refugia[16], a concept that provides a theoretical basis for the identification of species-specific safe areas under climate change. In that respect, targeted species translocation is sometimes advocated as a potential solution to safeguard species for conservation that would otherwise become extinct in the face of climate change[17,18]. While considerable controversy has emerged regarding the selection of geographic areas translocation[19], it is often suggested that SDM are an appropriate tool for selecting areas for translocations that are becoming suitable in the future[18]. Our results suggest that should SDMs be used for guiding conservation translocation, it should be done carefully, since SDMs tend to cause high levels of uncertainty. Running multiple state-of-the-art SDMs forced by several GCMs seems to be the best and only option to provide ensemble future projections for assessing options for translocations. Ensemble projections provide information what areas are suitable in most models and scenarios, which reduces translocation uncertainties compared to using projections from a single SDM with a single climate forcing only.

Alternatively, in situ conservation planning focuses on protecting or managing species where they currently occur and protect them from other, negative effects. To guide efforts to areas within the current range that are least affected by climate change, the use of multiple GCM and especially RCPs is very important.

Interestingly, when focusing on sensitivity metrics related to species loss or temporal turnover, the uncertainty related to variation in SDMs is not necessarily larger than in GCMs and is lower than in RCPs, which is as expected. When researchers do not have the computational capability to implement a full treatment of uncertainty, focusing on those metrics might be an avenue. In terms of conservation planning, it also means that optimization algorithms that rather focus on securing diversity (both α- and β-diversities in a complementary way) should be less affected by uncertainty from GCMs and SDMs. They should thus mostly concentrate their efforts on assessing the effects of emission scenarios. Alternatively, there are also alternative modelling techniques that explicitly focus on community-level metrics (i.e. pixel-based). Since those approaches do not rely on stacking individual SDM, they are less prone to bias coming from aggregating models with different quality[20]. However, modelling α- and β-diversity explicitly is not as straightforward than modelling individual species[21]. Similar analyses than the one proposed here but with community-level modelling approaches will be interesting to understand whether they are less influential on projection outputs than SDMs. In this study, we have also proposed novel ways of communicating both the uncertainties and their sources either per region, sub-region or pixel. This shall help pave the way to better communicate and map both the metric and the variance in this metric from the different sources of uncertainty. Yet, it shall also illustrate how the importance of these different sources varies in space and time and that conservation actions or biodiversity management should account for those variations. We have seen that sources of uncertainty can strongly vary in space depending on the quality of biodiversity models for some particular groups or the variability of projections among GCM (Supplementary Figs. 5–8).

Biodiversity scenarios are not meant to predict the future precisely, but rather to project the range of possible futures allowing to better understand uncertainties and alternative visions of this future. These visions allow for considering how different political options, represented here by the RCPs, might influence the persistence of biodiversity under a wide range of possible futures. However, to be useful, one has to acknowledge also other sources of variations that may influence the overall modelling exercise[12]. There are different types of biodiversity models, and here while considering only a single type (i.e. statistical SDM), we show that different algorithms could lead to substantial variability in the sensitivity metrics used. This variation may blur the utility of using several RCPs to discuss their impact on biodiversity persistence. This issue was long recognized in climate sciences such that it is no longer conceivable to show projected climatic trends from a single GCM only. Rather, ensembles of climate trajectories are usually shown (or least statistics thereof) and offered to users through data portals (e.g. CMIP5). The biodiversity modelling community needs to more consistently follow this path and report and communicate the variability resulting from the different options in biodiversity models (e.g. SDMs, dispersal scenarios) and input data (e.g. GCM, RCP). However, the range of biodiversity model types, algorithms, input data or parameterizations is so large that it seems currently impossible to report the variability even across biodiversity model types. However, we urge that variabilities originating from modelling algorithms, input data and external forces be assessed and reported comprehensively for better informing science, users and decision-makers in exploring options for the future.

## Methods

**Statistical analyses**. All analyses have been carried out in the R environment (specific functions within specific package are indicated in parentheses).

**Data**. We used the distribution maps provided by the Amphibian and Mammal Red List Assessment (http://www.iucnredlist.org/) for 5547 and 4616 species, respectively. For birds, breeding range distribution maps were extracted from BirdLife (http://www.birdlife.org/) for 9993 species. Ranges were converted to 100 km × 100 km equal-area grid cells, a resolution previously validated as the finest justifiable for these data globally[22]. Grid cells within the distribution range of each species were thus converted to presence points, while those outside their distribution ranges were converted as absence points. We finally focused on 1351 amphibian, 7248 bird and 2896 mammal species after removing species occurring in <20 grid cells, as well as domestic and aquatic species. We consider 20 presence points the minimum to successfully fit response curves to four different predictor variables.

**Climatic data**. Current climate (1979–2013) was represented by four bioclimatic variables from the CHELSA dataset[23] up-scaled from a 1 km to a 100 km resolution. The chosen variables were as follows: annual mean temperature, annual temperature range, annual sum of precipitation and precipitation seasonality (coefficient of variation in monthly sum of precipitations).

Projected future climate variables were taken from five GCMs driven by four scenarios of RCPs in a factorial manner as explained in Supplementary Fig. 1. The five selected models originate from the CMIP5 collection of model runs used in IPCC's 5th Assessment Report (IPCC 2013). The five models from which data were taken are: CESM1-BGC[24] run by National Center for Atmospheric Research (NCAR); CMCC-CMS[25] run by the Centro Euro-Mediterraneo per i Cambiamenti Climatici (CMCC); CM5A-LR[26] run by the Institut Pierre-Simon Laplace (IPSL); MIROC5[27] run by the university of Tokyo; and ESM-MR[28] run by Max Planck Institute for Meteorology (MPI-M).

Future climatic conditions of the four climatic variables were also taken from the CHELSA dataset[23], which provides CMIP5 scenarios at a native resolution of 30 arc seconds. Future conditions from coarser resolution GCMs had been achieved using climatologically aided interpolation. We took the difference between selected GCMs from CMIP5 at a 0.25° grid cell size for current conditions

(1979–2013) and the selected future periods (2041–2060, 2061–2080) and interpolated them using b-spline interpolation to the resolution of 30 arc seconds of CHELSA. The resulting difference was then added to (for temperature) or multiplied with (for precipitation) to the CHELSA climatologies of the 1979–2013 baseline period. As our study used 100 km grid cells, the native CHELSA resolution was upscaled by calculating mean within values per each 100 km grid cell.

**Species distribution models**. An ensemble of projections of SDM was obtained for the 11,495 selected species. The ensemble included projections with Generalized Additive Models, Boosting Regression Trees, Generalized Linear Models and Random Forests. Models were calibrated for the baseline period using 70% of observations randomly sampled from the initial data and evaluated against the remaining 30% data using the true skill statistic (TSS[29]). Presence data were randomly drawn from the gridded range maps. For absences, we considered data in a reasonable buffer around the presence data to avoid having over-optimistic predictive accuracies[30]. To be consistent with assumed realistic dispersal distances (see next paragraph) and in line with previous analyses, we selected absence data in 2000, 3000, and 4000 km buffer around amphibian, mammal and bird species ranges, respectively. This analysis was repeated four times, thus providing a four-fold internal cross-validation of the models (biomod package[15]). The quality of the models was 'very high' to 'excellent' with an average TSS of 0.83 (Supplementary Fig. 2). Since the quality of the models strongly affect projection uncertainties, we tested three thresholds below which models were removed from projection ensembles. We used a threshold of TSS = 0.4, which is usually considered a minimum for retaining reliable models[29]. We also used thresholds of 0.6 and 0.7, and investigated their effects on uncertainties of species-based sensitivity metrics. For all further analyses, the most drastic threshold (TSS = 0.7) was then used (red lines in Supplementary Fig. 2).

For each species, all calibrated models (4 SDMs × 4 repetitions) were then used to project the potential distribution of each species under both current and projected future climatic conditions (Supplementary Fig. 1).

**Dispersal limitation**. Since most species have a sub-global distribution, we adjusted the area from which species are modelled and for which projections are made. In other words, for amphibian, bird and mammal species, the modelled and projected area included all grid cells within 2000, 3000 and 4000 km of species' current distributions, respectively. This represents a maximal dispersal distance and excludes regions and climatic conditions that are outside of what is conceivably within reach for these species[30]. These estimates likely underestimate the true dispersal limitation of most species but give a more reliable estimate than assuming unlimited dispersal during this century. For most analyses, we also assumed a 'no dispersal' scenario. This 'no dispersal' scenario is useful to investigate whether and where area of currently suitable climate habitat will remain to be suitable for species in the future. This is a very important aspect for in situ conservation.

**Sensitivity metrics**. All species projections under future conditions (5,149,760 in total) were converted to a metric of species sensitivity. CCS measures the relative change in climatic suitability. It corresponds to the difference between the total suitable climatic area projected into the future under the assumption of limited dispersal and the total suitable area projected under current conditions, with the resulting quantity being divided by the total suitable area projected on the current conditions. Under the 'no dispersal' hypothesis, we derived LCS, which measure the relative loss in climatic suitability. This metric quantifies a species' risk of habitat loss within its current area of occupancy.

At the pixel-level, we calculated several metrics commonly used in biodiversity scenario modelling. First, we calculated the relative (percent) change in species richness (Δα-diversity) and the relative (percent) loss of species per pixel (% loss). Second, we calculated the temporal species turnover per pixel under the assumption of limited dispersal (βt = [No. of species lost + No. of species gained]/ [current species richness + No. of species gained]). Finally, to be useful for the global assessment of the IPBES, we also estimated the change in spatial turnover (Δβs) within each of the IPBES-subregions (Supplementary Fig. 16), calculated as the relative (percent) change in total diversity per sub-region (γ-diversity) divided by the mean α-diversity per sub-region. Δβs was calculated under both no dispersal and limited dispersal assumptions.

**Variance partitioning of the uncertainty**. For both species-based sensitivity metrics and pixel-based variation in community metrics, we conducted a set of variance partitioning to understand the main drivers of the variance.

All sensitivity metrics are based on a large amount of simulations (e.g. 448 projections per species) that vary as a result cross-validation sub-setting of initial data, dispersal scenario, and choice of SDM, GCM, and RCP. First, the effects of cross-validation explained <1% of total variation and thus we decided not to consider it for further analyses. Cross-validated models were considered as four independent runs of the same models. Second, since our sensitivity metrics were defined for different dispersal assumptions, we did not consider dispersal in the variance partitioning, but contrasted the results as a function of it. Finally, we partitioned the effects of SDMs, GCMs and RCPs on the final metrics using a nested ANOVA, in which SDMs were the first level, followed by GCMs and RCPs,

which were considered as crossed effects (SDM/GCM:RCP). We implemented a nested ANOVA since SDMs are first fitted irrespective of GCMs and RCPs, yet they differ strongly in how they affect projected suitable habitats when applied to GCMs and RCPs. Therefore, we considered the effects from GCM and RCP as nested within the effects of SDMs. We are aware that most analyses have been done with a full factorial (non-nested) design so far. For the sake of consistency, we also performed a full-factorial ANOVA that showed the same results. Since we believe the nested ANOVA is more correct, we kept it as main effect in this study. From the nested ANOVA, we focused on the deviance explained by each component.

For the species-level sensitivity metrics, we evaluated several selection criteria below which SDMs were not retained for final ensemble projections (see Species distribution models part). Here, the nested ANOVAs were performed for three TSS thresholds (0.4, 0.6 and 0.7) and the results were compared to assess whether the explained variance that come from SDMs is driven by the quality of retained SDMs. Since we observed increasing variability caused by SDMs when using too low thresholds, we kept the highest threshold TSS (TSS = 0.7) to ensures that only very good models were retained.

**Reporting Summary**. Further information on experimental design is available in the Nature Research Reporting Summary linked to this article.

## Data availability

All data used in this paper are freely available and downloadable from the web. Species distribution maps were provided by the Amphibian and Mammal Red List Assessment (http://www.iucnredlist.org/). For birds, breeding range distribution maps were extracted from BirdLife (http://www.birdlife.org/). All climatic data are available on the CHELSA data portal (http://chelsa-climate.org).

## Code availability

The R code for running the entire analysis is available on https://gricad-gitlab.univ-grenoble-alpes.fr/leca/publications/thuiller_2019_natcomm.

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

## Acknowledgements

W.T. thanks the Technical Support Unit of the IPBES for the Expert Group on Models and Scenarios for funding. We also thank H.M. Pereira, R. Alkemade and P.W. Leadley for their support and valuable suggestions. W.T., D.N.K. and N.E.Z. received funding from the ERA-Net BiodivERsA - Belmont Forum, with the national funder Agence Nationale pour la Recherche (ANR-18-EBI4–0009) and Swiss National Foundation (20BD21_184131/1), part of the 2018 Joint call BiodivERsA-Belmont Forum call (project 'FutureWeb'). N.E.Z. & W.T. further acknowledge support from the SNF/ANR grant 310030L-170059 / ANR-16-CE93-004. As part of the CDP-Trajectories project, this work was also supported by the Agence Nationale pour la Recherche in the framework of the "Investissements d'avenir" program (ANR-15-IDEX-02).

## Author contributions

W.T. conceived and planned the analyses, J.R. prepared the species distribution data, D.K. and N.E.Z. prepared the climatic layers, M.G. ran the models and the uncertainty analyses, W.T. and M.G. conceived the figures and outputs, W.T. wrote the initial draft of the paper and all authors contributed to subsequent revisions.

## Additional information

**Competing interests:** The authors declare no competing interests.

