## [Peer Review File · Nature Communications]

Reviewers' comments:

Reviewer #1 (Remarks to the Author):

This manuscript partitions the sources of variation in projections of biodiversity change to evaluate how model algorithm, climate model, and emissions scenario contribute to uncertainty. As the authors note, their effort in developing alternative biodiversity projections is the largest to date, and I agree that this gives it value. However, I still have reservations and questions about the paper, both about the modeling and the implications for policy.

In the response letter, the authors highlight three reasons their manuscript is important and novel. The first reason is spatial coverage and uncertainty quantification; I agree that this manuscript is more comprehensive in scope than preceding studies and would be useful in that respect. I am less confident in the generality of the results they show in terms of the second reason for novelty: 'differential effects across different metrics'. By this they mean that SDMs were the only important source of uncertainty for some responses, while RCP and GCM mattered for others. There is no question that different statistical algorithms can produce quite different future projections, even when mapped suitability in the historical/current time period appears similar; many previous studies have focused on sensitivity to model algorithm. However, it is not clear to me why SDM would be the only major source of explained variation in projections that allowed for both range gain and loss, while RCP and GCM also mattered when just considering range loss. Is there that much more extrapolation in climate space when considering a buffer around the occupied range? And why wouldn't the RCP/GCM also affect how much extrapolation there is (e.g., scenarios and models projecting more change lead to greater extrapolation)? It feels to me that the authors report the results of the variance partitioning, but do not try to understand them by exploring how they arose. I therefore have a hard time buying into their argument for the central finding with respect to policy and conservation strategy: that studies focused on conserving existing populations should consider multiple RCPs, while those interested in potential future habitat outside the current range should mostly focus on variation among statistical algorithms. Why should this be the case? How might it be affected by the spatial grain of the analysis, i.e., might a study with smaller grid cells find different results here? Obviously the authors can't explore the spatial grain, but my point is that without understanding why this finding arises it isn't clear whether or not it will be robust.

The third point of novelty the authors emphasize is uncertainty over space and as a function of the number of occurrences. Much of this material has been added in the revision in response to the second reviewer, and I agree that it adds value, particularly the figure on how the contributions of SDM, GCM, and RCP to the uncertainty vary with range size. Overall, the concerns of the second reviewer were carefully addressed.

The basic concerns about the modeling methodology remain present. Inferring absences from buffers outside range polygons is likely to overestimate model performance. Spatial cross-validation could be one way to begin to assess performance more realistically. Nevertheless, I agree that the methods used follow those in the literature, and are in line with SDM best practices for these data inputs. Moreover, SDMs are the only tool that can produce spatially-explicit projections for species with limited data (i.e., almost all species). The question is whether SDMs based on these low quality data produce actionable information for conservation or policy.

Minor comments:

Page 1: 'these models project future biodiversity trajectories'; usually the future state is projected, not the whole trajectory

Page 2, and throughout: Previous studies have suggested the importance of RCP depends on the time period of projection (e.g., Snover et al. 2013 *Con Bio*). It is surprising that RCP is more important than GCM in 2041-2060 (whereas it is expected for the later period). In mid-century,

there is as much variation in the climate projections among GCMs as among RCPs (e.g., in IPCC plots). Where do these 5 GCMs fall amongst other climate models in the amount of change they project in the 4 selected climate variables? One might even ask whether the findings in Fig 4 align with those from climate models themselves – i.e., are there places in the world with relatively more variation among RCPs than among GCMs?

Page 2, and throughout: I think you should define and use 'sensitivity metric' carefully, given that sensitivity in the SDM literature can also refer to the probability of predicting presence given that a location is occupied.

Page 9 SDM section: 400km should be 4000

Page 10: The variance partitioning section could be clearer. You might introduce that section by explaining that analyses were done to understand variation in species-level vulnerability metrics better (Fig 1-3), and separate analyses were done to understand pixel-level variation in community metrics (Fig 4).

All figures: The figures in general are quite visually appealing.

Fig 3: Explain the y-axis better. I thought these bar plots showed the percentage of explained variation, so I'm not sure why for amphibians it can be over 100? Is the Total bar for the proportion of all variance that was explained by all things in the model?

Fig 4: Explain the inset/legend more clearly. Does the gray shading show the global mean for each variable? Why is it gray? The color corresponds to the proportion of explained variance in the model attributable to each factor?

Reviewers Comments:

Reviewer #1 (Remarks to the Author):

This manuscript partitions the sources of variation in projections of biodiversity change to evaluate how model algorithm, climate model, and emissions scenario contribute to uncertainty. As the authors note, their effort in developing alternative biodiversity projections is the largest to date, and I agree that this gives it value. However, I still have reservations and questions about the paper, both about the modeling and the implications for policy.

In the response letter, the authors highlight three reasons their manuscript is important and novel. The first reason is spatial coverage and uncertainty quantification; I agree that this manuscript is more comprehensive in scope than preceding studies and would be useful in that respect. I am less confident in the generality of the results they show in terms of the second reason for novelty: ‘differential effects across different metrics’. By this they mean that SDMs were the only important source of uncertainty for some responses, while RCP and GCM mattered for others. There is no question that different statistical algorithms can produce quite different future projections, even when mapped suitability in the historical/current time period appears similar; many previous studies have focused on sensitivity to model algorithm. However, it is not clear to me why SDM would be the only major source of explained variation in projections that allowed for both range gain and loss, while RCP and GCM also mattered when just considering range loss. Is there that much more extrapolation in climate space when considering a buffer around the occupied range? And why wouldn’t the RCP/GCM also affect how much extrapolation there is (e.g., scenarios and models projecting more change lead to greater extrapolation)?

We thank the referee for his/her very insightful comments. These are fair points.

We believe this is not only a matter of whether RCP/GCM do not affect extrapolation, they do, but rather that the difference between some of the simple models used here (GLM and GAM) in respect to the more complex models (RF and BRT) overrides the effect of RCP/GCM. Complex SDM can lead to different combinations of features producing similar model performance in the present (Maggini et al. 2006), but vastly diverging spatial predictions when transferred to other conditions (Thuiller 2003, Thuiller et al. 2004, Pearson et al. 2006, Edwards et al. 2006, Elith et al. 2010).

It feels to me that the authors report the results of the variance partitioning, but do not try to understand them by exploring how they arose. I therefore have a hard time buying into their argument for the central finding with respect to policy and conservation strategy: that studies focused on conserving existing populations should consider multiple RCPs, while those interested in potential future habitat outside the current range should mostly focus on variation among statistical algorithms.

We disagree, we did understand and explain the results of the variance partitioning. We have now added some more text to explain it. Perhaps we were not clear enough. We have now modified Supp Figure 3 to exemplify why SDMs do explain much of the variance in respect to RCP and GCM. It completely makes sense since this is not only a matter of extrapolation (where indeed RCP should also be important), but also a crucial matter of how simple or complex are the estimated relationships between the presence-absence of species and the environmental variables.

Finally, we have removed some of the recommendations that could indeed be too strongly interpreted.

Why should this be the case? How might it be affected by the spatial grain of the analysis, i.e., might a study with smaller grid cells find different results here? Obviously the authors can’t explore the spatial grain, but my point is that without understanding why this finding arises it isn’t clear whether or not it will be robust.

We do not believe the spatial grain will matter much, since then climate models and their downscaling versions are of poor quality too. So perhaps, the effects of GCM will be stronger but this might only come from poor climate data. Difficult to say indeed.

We hope now that we the new Supp Figure 3, it demonstrates the over-riding of SDM in one case and not in the other one.

The third point of novelty the authors emphasize is uncertainty over space and as a function of the number of occurrences. Much of this material has been added in the revision in response to the second reviewer, and I agree that it adds value, particularly the figure on how the contributions of SDM, GCM, and RCP to the uncertainty vary with range size. Overall, the concerns of the second reviewer were carefully addressed.

Thanks a lot.

The basic concerns about the modeling methodology remain present. Inferring absences from buffers outside range polygons is likely to overestimate model performance. Spatial cross-validation could be one way to begin to assess performance more realistically. Nevertheless, I agree that the methods used follow those in the literature, and are in line with SDM best practices for these data inputs. Moreover, SDMs are the only tool that can produce spatially-explicit projections for species with limited data (i.e., almost all species). The question is whether SDMs based on these low quality data produce actionable information for conservation or policy.

We agree on that point, but there is nothing we can do about it. We note that the overall goal of the paper is not to give actionable information for conservation or policy but rather give guideline on how to interpret results from global biodiversity scenarios. We town down the policy aspect of the paper as recommended;

Minor comments:

Page 1: 'these models project future biodiversity trajectories'; usually the future state is projected, not the whole trajectory

Thanks. We changed as: these models project future biodiversity patterns.

Page 2, and throughout: Previous studies have suggested the importance of RCP depends on the time period of projection (e.g., Snover et al. 2013 Con Bio). It is surprising that RCP is more important than GCM in 2041-2060 (whereas it is expected for the later period). In mid-century, there is as much variation in the climate projections among GCMs as among RCPs (e.g., in IPCC plots). Where do these 5 GCMs fall amongst other climate models in the amount of change they project in the 4 selected climate variables? One might even ask whether the findings in Fig 4 align with those from climate models themselves – i.e., are there places in the world with relatively more variation among RCPs than among GCMs?

Interesting indeed. Here, we critically selected the GCM to be the ones summarizing the overall diversity across all available GCMs. In Figure 4 of Sanderson et al. 2015 (J. of Climatology), they highlighted the set of models that account for most of the variance in global earth climate (see <https://journals.ametsoc.org/doi/10.1175/JCLI-D-14-00362.1>). Our selected GCMs are the most important ones (they should be selected from right to left).

Now, we disagree that RCPs should mostly be important for 2080. This is was relatively true for the former SRES scenarios (as the paper cited by the referee), but the RCPs are strikingly different for 2050 already, and there is no expected a priori that GCM should lead higher variance in 2050 than RCP and the way around for 2080. That was true for SRES scenario, but this is not the case anymore.

Page 2, and throughout: I think you should define and use 'sensitivity metric' carefully, given that sensitivity in the SDM literature can also refer to the probability of predicting presence given that a location is occupied.

Good point. We rewrote the part introducing the different sensitivity metrics as suggested.

Page 9 SDM section: 400km should be 4000

Changed.

Page 10: The variance partitioning section could be clearer. You might introduce that section by explaining that analyses were done to understand variation in species-level vulnerability metrics better (Fig 1-3), and separate analyses were done to understand pixel-level variation in community metrics (Fig 4).

Changed as suggested.

All figures: The figures in general are quite visually appealing.

Thanks!

Fig 3: Explain the y-axis better. I thought these bar plots showed the percentage of explained variation, so I'm not sure why for amphibians it can be over 100?

We modified the legend, there was indeed a mistake. This is not the percentage of explained variation but rather the absolute explained deviance. This is why Amphibians can go over 200. Transforming the absolute values in percentages lead to unreadable bars given the large differences between the deviance explained by the different components.

Is the Total bar for the proportion of all variance that was explained by all things in the model?

Yes; We added this important information.

Fig 4: Explain the inset/legend more clearly. Does the gray shading show the global mean for each variable? Why is it gray? The color corresponds to the proportion of explained variance in the model attributable to each factor?

That was indeed misleading. We have now changed the colors scheme for the insert, it is only a single grey color.

REVIEWERS' COMMENTS:

Reviewer #1 (Remarks to the Author):

The authors evaluate how SDM algorithm, RCP, and GCM contribute to uncertainty in estimates of range loss and change in range size. As noted earlier, the scope of their paper is comprehensive, but I have concerns about the utility of these type of SDMs for informing management or policy. The authors note in their response that the overall goal is not to produce actionable information, but to guide the allocation of effort when SDM studies cannot consider all options. It's too bad the field is at a place where guiding sensitivity analyses for perhaps questionable models is a contribution; unfortunately, I do agree with the authors that the field is still there and we lack other practical tools or a more reliable way to assess extrapolation performance (not based on random cross-validation).

For pixel-based measures of sensitivity, it would be good to acknowledge somewhere the literature on the merits of stacking independent SDMs versus modeling community-level metrics, as this is another source of uncertainty.

Line 24: perhaps 'greatest use', rather than 'any use'

Line 103: should be 'away'

Line 288: all species

Fig 1: the shading for TSS scale is not clear, or the lighter bars aren't showing up? It is striking how little of the explained deviance is attributable to these factors.

S3: The x-axis on the figure itself doesn't correspond to what is in the legend; please fix.

REVIEWERS' COMMENTS:

Reviewer #1 (Remarks to the Author):

The authors evaluate how SDM algorithm, RCP, and GCM contribute to uncertainty in estimates of range loss and change in range size. As noted earlier, the scope of their paper is comprehensive, but I have concerns about the utility of these type of SDMs for informing management or policy. The authors note in their response that the overall goal is not to produce actionable information, but to guide the allocation of effort when SDM studies cannot consider all options. It's too bad the field is at a place where guiding sensitivity analyses for perhaps questionable models is a contribution; unfortunately, I do agree with the authors that the field is still there and we lack other practical tools or a more reliable way to assess extrapolation performance (not based on random cross-validation).

We thank the referee for his/her very helpful comments and suggestions during the whole revision process.

For pixel-based measures of sensitivity, it would be good to acknowledge somewhere the literature on the merits of stacking independent SDMs versus modeling community-level metrics, as this is another source of uncertainty.

Good point. We have added few sentences in the discussion around this issue.

Line 24: perhaps 'greatest use', rather than 'any use'

Changed.

Line 103: should be 'away'

Changed.

Line 288: all species

Changed

Fig 1: the shading for TSS scale is not clear, or the lighter bars aren't showing up? It is striking how little of the explained deviance is attributable to these factors.

We modified Figure 1 to be clearer.

S3: The x-axis on the figure itself doesn't correspond to what is in the legend; please fix.

Indeed, we modified the caption.